# Targeting HER2-Positive Solid Tumors with CAR NK Cells: CD44 Expression Is a Critical Modulator of HER2-Specific CAR NK Cell Efficacy

**DOI:** 10.3390/cancers17050731

**Published:** 2025-02-21

**Authors:** Bence Gergely, Márk A. Vereb, István Rebenku, György Vereb, Árpád Szöőr

**Affiliations:** 1Department of Biophysics and Cell Biology, Faculty of Medicine, University of Debrecen, 4032 Debrecen, Hungary; 2HUN-REN-UD Cell Biology and Signaling Research Group, Faculty of Medicine, University of Debrecen, 4032 Debrecen, Hungary; 3Faculty of Pharmacy, University of Debrecen, 4032 Debrecen, Hungary

**Keywords:** CAR NK cell, HER2-CAR, trastuzumab, solid tumor, extracellular matrix, CD44

## Abstract

Natural killer (NK) equipped with a chimeric antigen receptor (CAR) targeting HER2 can offer a novel treatment for HER2-positive solid tumors, which are often resistant to conventional monoclonal antibody therapies like trastuzumab. This study investigates the efficacy of first, second, and third-generation HER2-CAR NK cells in targeting both trastuzumab-sensitive and trastuzumab-resistant HER2-positive tumors, with a particular focus on the role of CD44 expression in modulating treatment outcomes. By comparing the performance of various CAR constructs, this study seeks to identify strategies that enhance the penetration and effectiveness of CAR NK cells in the challenging tumor microenvironment, leading to improved therapeutic options for patients with resistant cancers, potentially offering a safer and more accessible alternative to CAR T cell therapies.

## 1. Introduction

Monoclonal antibody therapies have revolutionized the treatment of various cancers by targeting specific tumor antigens, leading to improved patient outcomes [1]. However, resistance to these therapies remains a pervasive challenge across many tumor types [2]. This resistance often arises from changes within the tumor microenvironment, such as the overexpression of extracellular matrix components that can shield tumor antigens from antibody binding [3]. Such mechanisms not only hinder the effectiveness of monoclonal antibodies but are also correlated with a poor prognosis [4]. Among these components, the role of CD44 is particularly critical. CD44 facilitates the tumor cells’ interaction with the extracellular matrix, enhancing the physical barrier around tumor cells and contributing to an immunosuppressive microenvironment. The interaction between CD44 and hyaluronic acid, a major component of the extracellular matrix, further stabilizes this protective niche, complicating the accessibility of targeted therapies to tumor antigens [5,6].

To combat this issue, innovative therapeutic strategies are being explored. Among these, chimeric antigen receptor (CAR) T cell therapy has shown significant promise [7]. Our previous studies demonstrated that HER2-specific CAR T cells could effectively eliminate clinic-derived and in vitro-generated HER2-targeting monoclonal antibody (trastuzumab)-resistant tumors. In contrast, the co-administration of trastuzumab with a CD16.176V.NK-92 transgenic natural killer (NK) cell that was designed to enhance antibody-dependent cellular cytotoxicity (ADCC) did not achieve similar results [8,9], indicating that CAR-modified immune cells can potentially be effective treatments for cancers resistant to mAb therapies.

Despite their promising potential, the use of CAR T cells in clinical settings is constrained by significant side effects and complex production processes [10]. These challenges highlight the need for alternative approaches that can leverage the precision of CAR technology while reducing associated risks and complexities. Recent preclinical trials suggest that natural killer (NK) cells, integral components of the innate immune system, could be effective substitutes since these effectors offer several advantages over T cells in CAR therapies. NK cells are less likely to induce Graft versus Host Disease (GvHD) compared to T cells, making them safer for allogeneic transplantations [11]. Moreover, they can rapidly recognize and kill target cells without prior sensitization, which is beneficial for immediate responses against cancer cells [12]. By engineering NK cells to target specific tumor antigens using chimeric antigen receptors, it might be possible to retain the targeted killing capacity of CAR T cells but with potentially fewer complications and a streamlined production process [13]. CAR NK cells hold the potential to be used as off-the-shelf products, reducing the time and complexity of the therapy as they do not require individual patient customization like CAR T cells [14].

Recent summaries of clinical trials that utilize CAR NK cells to target solid tumors collectively emphasize the encouraging therapeutic potential of these effector cells in treating solid tumors; however, they also point out the major challenges that must be addressed [15,16]. First and foremost, no conclusive evidence supports their therapeutic efficacy in monoclonal antibody-resistant tumors. The complexity of tumor microenvironments, especially those that have evolved resistance to standard mAb therapies, presents a challenging landscape for CAR NK cells [17]. The mechanisms through which these cells recognize, engage, and destroy mAb-resistant tumor cells are poorly understood, and thus it is still unknown whether CAR NK cells can identify and destroy resistant tumors like CAR T cells or if they fail to be effective, similarly to ADCC, due to an inability to access antigens. Furthermore, the optimal CAR construct that would elicit the most effective CAR NK cell effector response in such contexts is still under investigation. However, defining the most effective NK-specific CAR backbone—considering factors such as the choice of antigen targets, the configuration of signaling domains, and the incorporation of costimulatory elements—remains a complex endeavor. Each of these components can significantly influence the overall efficacy of CAR NK cells, impacting their ability to overcome the adaptive resistance mechanisms of solid tumors.

In this study, we investigated whether HER2-specific CAR NK cells display the same efficacy against both trastuzumab-sensitive (CD44^−^) and trastuzumab-resistant (CD44^+^) HER2^+^ tumor lesions. We modified NK-92 cells to express a HER2-specific CAR, incorporating CD28 and/or 41BB costimulatory domains and targeting the same epitope as trastuzumab. These CAR NK-92 cells showed comparable performance in vitro against both types of tumors grown in two-dimensional monolayers. However, while they were effective against CD44-negative N87 spheroids and corresponding in vivo xenografts, they failed to impact the spheroids and xenografts of the CD44-positive JIMT-1. This outcome underlines the challenges CAR NK cells face in penetrating the extracellular matrix (ECM) and overcoming its associated resistance in solid tumors. Furthermore, it reinforces the previously noted observation that T cells tend to proliferate more effectively than NK cells in the microenvironment of solid tumors.

## 2. Materials and Methods

All materials were purchased from Sigma-Aldrich (St. Louis, MO, USA) unless indicated otherwise.

### 2.1. Cells and Culture Conditions

HEK293T packaging cells, MDA-MB-468 (triple-negative human breast cancer cell line), and N87 (gastric cell line) cells were acquired from the American Type Culture Collection (Manassas, VA, USA). A cell line stably expressing HER2 (MDA-HER2) was generated from MDA-MB-468 [18]. These adherent cell lines were grown in DMEM (Dulbecco’s Modified Eagle Medium) medium with 10% Fetal Bovine Serum (FBS), 2 mM GlutaMAX, and antibiotics. The JIMT-1 human breast cancer cell line was established by and is a kind gift from Jorma Isola (Laboratory of Cancer Biology, University of Tampere, Finland) [19]. JIMT-1 cells were cultured in a 1:1 ratio of Ham’s F-12 and DMEM supplemented with 20% FBS, 300 U/L insulin, 2 mmol/L GlutaMAX, and antibiotics.

MDA.ffLuc, MDA-HER2.ffLuc, N87.ffLuc, and JIMT-1.ffLuc were generated by the single-cell cloning of MDA-MB-468, MDA-HER2, N87, and JIMT-1 cell lines following their transduction with a retrovirus encoding eGFP.ffLuc to express an enhanced green fluorescent protein (eGFP) and firefly luciferase (ffLuc).

CD16.176V.NK-92 cells (referred to as NK-92), kindly provided by Dr. Kerry S. Campbell, the Fox Chase Cancer Center, Philadelphia, PA, on behalf of Brink Biologics, Inc., San Diego, CA, were used as effectors. This line has been produced by transducing the parental NK-92 cell line (ATCC CRL-2407, which lacks endogenous expression of CD16 [20]) to express the high-affinity variant of CD16 (FcγRIIIA 176V; see VAR_003960 entry within P08637 and BC017865.1) [21,22]. The cells were cultured in a complete NK medium of α-MEM containing 10% FCS and 10% horse serum, supplemented with glutamine, non-essential amino acids, Na-pyruvate, antibiotics, and IL-2 at 400 IU/mL (Proleukin S, Clinigen Ltd., Burton-on-Trent, UK).

In ADCC experiments, MDA, MDA-HER2, N87, and JIMT-1 cells were cultured in a complete NK medium.

All the cells and cell lines were maintained in a humidified atmosphere containing 5% CO_2_ at 37 °C and were routinely checked for the absence of mycoplasma contamination.

### 2.2. Retrovirus Production and Transduction of NK-92 Cells

Retroviral particles were produced by transiently transfecting HEK 293T cells with HER2-CAR-encoding pSFG retroviral vectors [23,24], along with the Peg-Pam-e plasmid encoding MoMLV gag-pol and the pMax.RD114 plasmid-encoding RD114, using the jetPrime transfection reagent (Polyplus, Illkirch, France). The HER2-specific chimeric antigen receptor constructs were designed with a backbone consisting of the IgG heavy chain signal peptide, the HER2-specific single-chain variable fragment FRP5, the IgG1 short hinge, and a human CD28 transmembrane region. The intracellular domain varied among constructs, including no costimulatory endodomain (.z), CD28 alone (CD28.z), 41BB alone (41BB.z), or a combination of CD28 and 41BB (CD28.41BB.z), all fused to the human CD3 zeta cytoplasmic domain [23,24]. Supernatants containing the retrovirus were harvested after 48 h.

To produce HER2-specific CAR NK-92 cells, 5 × 10^5^/mL NK-92 cells were pre-stimulated with 400 IU/mL IL-2 (Proleukin S, Clinigen) in a 24-well non-tissue culture plate in 2 mL complete NK-medium. NK cells were transduced with pseudoviral particles on 20 μg/mL RetroNectin-coated (Takara Bio, Kusatsu, Japan) plates in the presence of IL-2 (400 IU/mL). Non-transduced NK-92 cells expanded on RetroNectin-coated plate supplied with IL-2 (400 IU/mL) in complete NK-medium. Following a 48 h incubation period, the genetically modified NK-92 cells were harvested from the transduction plate and cultured in a complete NK-medium supplemented with 400 IU/mL IL-2 for seven days. After this expansion period, all HER2-CAR NK-92 cell lines underwent flow cytometric sorting, achieving a purity level exceeding 90% in all cases.

### 2.3. Flow Cytometry

To verify HER2-CAR expression, cells were directly labeled with 5 µg/mL Alexa Fluor 647-conjugated monomeric HER2 extracellular domain (HER2 ECD; Cat.#10004-HCCH, Sino Biological Europe GmbH, Eschborn, Germany) for 10 min on ice. Flow cytometric analysis was conducted using a NovoCyte RYB flow cytometer, and data were processed with NovoExpress software v1.2.1. (Agilent Technologies, Santa Clara, CA, USA) for at least 10,000 cells per sample.

### 2.4. Cytokine Secretion Assay

A total of 100,000 HER2-specific CAR-transduced or non-transduced (NT) NK-92 cells (in the presence of 10 µg/mL trastuzumab) were plated onto 1 µg/mL HER2-Fc (Sino Biological Inc., Beijing, China) pre-coated plates or cocultured with MDA-HER2, JIMT-1, and N87 HER2^+^ target cells at a 1:1 effector to target ratio (E:T ratio). Following 24 h of culture, the supernatant was harvested and analyzed for the presence of interferon-gamma (IFNγ) by ELISA (R&D systems, Minneapolis, MN, USA) according to the manufacturer’s instructions using a Spark^®^ multimode microplate reader (Tecan Group Ltd., Männedorf, Switzerland). The HER2^−^ MDA-MB-468 cell line and NT NK-92 cells in the absence of trastuzumab served as negative target and effector controls, respectively.

### 2.5. Cytotoxicity Assay

The cytotoxic activity of CAR NK-92 cells against targets was determined using a luciferase-based cytotoxicity assay. MDA-MB-468, MDA-HER2, JIMT-1, and N87 cells expressing eGFP/ffLuc were plated in 96-well flat-bottom plates at a concentration of 3 × 10^4^ cells/well in triplicates. After 4 h, HER2-CAR NK-92 cells or NT NK-92 cells (in the presence of 10 µg/mL trastuzumab) were added at a 1:1, 0.3:1, 0.1:1, 0.03:1, 0.01:1, or 0.003:1 effector-to-tumor cell ratio in complete NK-medium. Wells without effector cells served as untreated controls. After 24 h, luciferase activity was determined using a luciferase assay kit according to the manufacturer’s instructions (Promega, Madison, WI, USA) and a Spark^®^ multimode microplate reader (Tecan Group Ltd., Männedorf, Switzerland). Empty media, NT NK-92 cells (in the absence of trastuzumab), HER2^+^ target cells cultured without the presence of effector cells, and the HER2^−^ MDA-MB-468.ffLuc cell line served as controls.

### 2.6. Proliferation Assay

HER2-CAR NK-92 cells were cultured in 96-well flat bottom plates at a concentration of 1 × 10^4^ cells/well in duplicates in the presence of 400 IU/mL IL-2. Every 3.5 days, effector cell number was determined by flow cytometry. At the same time, we calculated the proliferation rate by dividing the total number of cells at the end by the total number plated in the beginning of the 3.5-day period.

### 2.7. Three-Dimensional Cell Culture and Propidium Iodide Incorporation Assay

N87.ffLuc and JIMT-1.ffLuc cells (1 × 10^5^ cells/mL in 200 μL) were seeded onto 96-well U-bottom plates containing cold medium with 2.5% Matrigel (BD Biosciences, San Jose, CA, USA). The cell suspensions were then centrifuged at 1000× *g* for 10 min at 10 °C, and the resulting cell pellet was cultured for 10 days to allow for spheroid formation. The cytotoxic activity of HER2-CAR NK-92 and NT NK-92 cells, in the presence or absence of 10 µg/mL trastuzumab, was assessed in these three-dimensional (3D) cultures using a propidium iodide incorporation assay [8,25,26]. Spheroids of uniform size were co-cultured with 2 × 10^5^ effector cells, and after 24 h, the 3D cocultures were labeled with 1 µg/mL propidium iodide. Target cell killing was analyzed using a Zeiss LSM 880 confocal microscope.

### 2.8. Xenograft Tumors and In Vivo Treatment

NSG (NOD.Cg-Prkdc^scid/Il2rg^tm1Wjl/SzJ) mice were obtained from the Jackson Laboratory and maintained in a specific pathogen-free environment. All animal experiments adhered to FELASA guidelines and recommendations and complied with DIN EN ISO 9001 standards [27]. On day 0, each seven-week-old female NSG mouse received a subcutaneous injection in both flanks, with each injection containing 3 × 10^6^ N87.ffLuc or JIMT-1.ffLuc cells in 100 µL PBS, mixed with an equal volume of Matrigel (BD Biosciences, San Jose, CA, USA). Tumor growth was assessed using an IVIS Spectrum CT instrument (Perkin Elmer, Waltham, MA, USA). Prior to imaging, isoflurane-anesthetized mice were injected intraperitoneally (i.p.) with D-luciferin (150 mg/kg). Bioluminescence images were captured 10 min post-injection and analyzed using Living Image software Version 4.0 (Caliper Life Sciences, Waltham, MA, USA). Signal intensity measured as total photons per second per square centimeter per steradian (p/(s × cm^2^ × sr)) was obtained from identically sized ROIs. Effector cell-treated mice received on day 14 and then biweekly an intravenous (i.v.) dose of 5 × 10^6^ NT (NT group) or HER2-CAR NK-92 cells (CD28.z, 41BB.z and CD28.41BB.z groups). Mice co-treated with NT NK-92 cells and trastuzumab received on day 14 and then biweekly an i.v. dose of 5 × 10^6^ NT NK-92 cells, plus were treated with 100 µg trastuzumab in 100 µL PBS i.p. twice weekly from day 15 (NT + trastuzumab group) (Appendix A). All treated animals received 50,000 IU IL-2 in 100 µL PBS i.p. twice weekly from day 15. (Appendix A). Experiments were approved by the National Ethical Committee for Animal Research (# 5-1/2017/DEMÁB).

### 2.9. Statistical Analysis

GraphPad Prism 5 software (GraphPad Software, Inc., La Jolla, CA, USA) was used for statistical analysis. Data were presented as mean ± SD or SEM. For comparison between two groups, a two-tailed t-test was used. One-way ANOVA with Bonferroni’s post hoc test was used to compare three or more groups. Survival, measured from the time of tumor cell injection, was analyzed using the Kaplan–Meier method and log-rank test. *p* values < 0.05 were considered statistically significant.

## 3. Results

For investigating the efficacy of CAR NK cells against HER2-positive solid tumors, we used the clinic-derived JIMT-1 [19] and an in vitro generated MDA-MB-468 variant (MDA-HER2 for short), stably expressing ectopic HER2 [18] as trastuzumab-resistant (HER2^+^/CD44^+^) and the N87 cell line as trastuzumab-sensitive (HER2^+^/CD44^−^) tumor models [8]. MDA-MB-468 (MDA for short) served as the HER2^−^ control. Also, ffLUC expressing N87, JIMT-1, MDA, and MDA-HER2 variants was used where appropriate. As effector cells, we used and genetically modified the CD16.176V.NK-92 cells, a stably transfected variant of NK-92 [20] that expresses the high affinity (176 V) variant of the Fcγ receptor (CD16) [21,22].

### 3.1. Generation of HER2-Specific Human CAR NK Cells and Their Proliferation In Vitro

To compare the impact on tumor eradication of HER2-specific CAR NK cells as living drugs with the effect of NK cells targeted by passively diffusing trastuzumab, we generated human CAR NK-92 cell lines that stably express second and third-generation HER2-specific CARs. Our constructs consisted of an FRP5 scFv-based HER2-specific recognition domain, an IgG1 short hinge, a CD28 transmembrane domain, a CD3z effector endodomain (.z, first-generation), and either CD28 (CD28.z, second-generation), 41BB (41BB.z, second-generation), or both CD28 and 41BB (CD28.41BB.z, third-generation) costimulatory endodomains (Figure 1a). CD28 and 41BB are critical costimulatory molecules that enhance the effector functions of immune cells, notably T cells and NK cells. However, their roles and signaling patterns differ significantly, offering unique contributions to immune regulation. CD28 is predominantly a T cell-specific molecule that plays a foundational role in the initial activation and sustained response of T lymphocytes. [28]. In contrast, 41BB (CD137) serves as an important costimulatory molecule in both T and NK cells, enhancing their proliferation, survival, and effector functions [29] (Figure 1a). Following FACS sorting, all HER2-CAR NK cell products showed higher than 90% HER2-CAR positivity as judged by flow cytometry (Figure 1b).

First, we performed a proliferation assay to evaluate the impact of the various costimulatory endodomains on the expansion of NK cells (Figure 1c). The proliferative response was measured in the presence of 400 IU/mL IL-2. Our findings indicate that constructs containing the 41BB cytosolic endodomain induced higher proliferative activity than constructs expressing no or CD28 costimulatory endodomain alone. Notably, NK cells expressing the CD28.41BB.z CAR showed the most pronounced proliferative activity (Appendix A).

### 3.2. HER2-Specific CAR NK Cells Successfully Recognize and Kill Trastuzumab-Resistant and Sensitive HER2-Positive Target Cells in Monolayer Cultures

To investigate the in vitro HER2-CAR NK cell activation, we measured the release of IFNγ and cytotoxicity towards target cells as key effector functions. Incubation with HER2-Fc recombinant chimera proteins (Appendix A) or with HER2^+^ target cells (MDA-HER2, JIMT-1, or N87) induced IFNγ release by all types of HER2-specific CAR NK cells, indicating NK cell activation (Figure 2a, MDA-HER2, JIMT-1, N87). In cocultures with trastuzumab-resistant, CD44^+^ MDA-HER2, and JIMT-1 cells, all types of HER2-CAR constructs induced a higher amount of IFNγ secretion than in trastuzumab-sensitive CD44^−^ N87 cocultures. CD28.41BB.z CAR NK cells secreted the highest amount of cytokine in all cocultures. In the presence of 10 μg/mL trastuzumab, NT NK cells also recognized immobilized HER2 molecules and secreted IFNγ (Appendix A). However, in cocultures with HER2^+^ target cells, a saturating dose of trastuzumab could not induce IFNγ release (Figure 2a, NT + TRAST). This discrepancy likely results from the high-density antigen presentation and crosslinking provided by immobilized HER2-Fc compared to the potential immune suppression and lower antigen density of HER2^+^ target cells, which can limit NK cell activation. There was no cytokine secretion in the absence of effector cells, target antigen (immobilized or membrane-bound), or trastuzumab, indicating that activation strictly depends on targeting the antigen.

Subsequent experiments were conducted to assess the cytotoxic capabilities of various HER2-CAR NK cell lines, both in the presence (Appendix A) and absence of IL-2 (Figure 2b). These experiments also included comparisons with 10 µg/mL trastuzumab-supplemented NT NK cell cultures to simulate antibody-dependent cellular cytotoxicity (ADCC) (Figure 2b and Appendix A, NK + TRAST). These results demonstrate that in the presence of 400 IU/mL IL-2, NT NK cells showed a significant innate cytotoxicity that was comparable with the cytotoxic efficiency of HER2-CAR and trastuzumab-treated NT NK cells (Appendix A, NT vs. MEDIA: *p* < 0.001). In the absence of IL-2, NT NK cells exhibited negligible cytotoxic activity, even at elevated E:T ratios (Figure 2b, NT). Under this condition, NK cells expressing 41BB.z CAR constructs exhibited the most substantial antitumor effects across all tested cell lines. However, these effects were not significantly greater than those mediated by other CAR constructs (Figure 2b). In cocultures with trastuzumab-resistant cell lines, HER2-specific CAR-modified NK cells displayed substantially greater efficacy than trastuzumab-targeted NT NK cells (Figure 2b, MDA-HER2 and JIMT-1). Notably, CAR constructs incorporating the 41BB costimulatory domain demonstrated pronounced cytotoxic effects at a 1:1 effector-to-target (E:T) ratio, even in the absence of membrane-bound HER2 targets (Figure 2b, MDA). This effect was not observed in cocultures with either .z or CD28.z CAR or trastuzumab-treated NK cells. Moreover, in the absence of IL-2, NT NK cells exhibited negligible cytotoxic activity, even at elevated E:T ratios (Figure 2b, NT).

### 3.3. 41BB-Containing HER2-Specific CAR-Modified NK Cells Selectively Destroy Three-Dimensional CD44^−^ N87 Tumor Spheroids but Spare CD44^+^ JIMT-1 Spheroids

Our previous study has confirmed that the CD44-enriched extracellular matrix of trastuzumab-resistant tumors, such as those derived from JIMT-1 cells, creates a formidable barrier that shields the core region of tumor spheroids from antibody access and severely restricts trastuzumab-mediated cytotoxicity by NK cells [8]. Conversely, conventional HER2-CAR T cells could penetrate the ECM barrier and effectively recognize the masked HER2 antigens within the spheroid’s core, initiating a robust cytotoxic attack [8]. Motivated by these findings, we investigated the antitumor capabilities of HER2-CAR-targeted NK cells in three-dimensional tumor spheroids (Figure 3a) originating from both trastuzumab-resistant (JIMT-1) and sensitive (N87) cell lines (Figure 3b).

In our experimental setup, JIMT-1 and N87 tumor spheroids were cocultured with HER2-CAR NK and trastuzumab-directed NT NK cells for 24 h. NT NK cells in the absence of trastuzumab served as controls. Using confocal microscopy, we observed the cytotoxic effects through the uptake of propidium iodide (PI), a marker of cell death (Figure 3a).

We revealed that CD44^+^ trastuzumab-resistant JIMT-1 spheroids were largely unaffected by HER2-CAR NK cells, exhibiting minimal PI uptake primarily in the outer layers of the 3D cultures (Figure 3b, JIMT-1, HER2-CAR groups). A slightly higher but still limited cytotoxic effect was observed in spheroids co-incubated with trastuzumab-directed NK cells, affecting only the peripheral cells (Figure 3b, JIMT-1, NT + TRAST). In stark contrast, the CD44^−^, trastuzumab-sensitive N87 spheroids were effectively targeted and disintegrated by all variants of HER2-CAR NK cells. Notably, NK cells engineered with 41BB.z and CD28.41BB.z CAR constructs demonstrated a particularly aggressive cytotoxic response, dismantling the spheroid structure within 24 h. Samples treated with trastuzumab-targeted NT NK cells and untreated control NT NK cells displayed negligible antitumor effects, underscoring the specificity and efficacy of the 41BB-containing CAR constructs in overcoming ECM-mediated protection in the sensitive tumor model.

### 3.4. HER2-CAR NK Cells Are Unable to Eliminate CD44^+^, Trastuzumab-Resistant JIMT-1 Xenografts In Vivo

To investigate whether the different tissue penetration capacities result in different efficacies in tumor elimination in vivo, we established subcutaneous HER2^+^/CD44^+^ JIMT-1.ffLuc xenografts in NSG mice (Appendix A and Figure 4a–c). To test the in vivo cytolytic function of HER2-CAR NK and trastuzumab-directed NT NK cells, 5 × 10^6^ effector cells were administered i.v. from day 14, biweekly. In trastuzumab-treated groups, 100 µg mAb was injected twice per week from day 14 after tumor cell inoculation (Appendix A). We found that neither various HER2-CAR NK cell treatments nor simultaneous administration of trastuzumab plus NK cells were able to induce regression of trastuzumab-resistant JIMT-1.ffLuc xenografts and improve overall survival (Figure 4a–c).

### 3.5. HER2-CAR NK Cells Exert Significant In Vivo Antitumor Effect Against CD44^−^, Trastuzumab-Sensitive N87 Xenografts

Finally, we evaluated the efficacy of HER2-targeting NK cells, comparing HER2-specific CAR NK cells and trastuzumab-redirected NK cells in a trastuzumab-sensitive xenograft model. We grew subcutaneous HER2^+^/CD44^−^ N87.ffLuc tumors in NSG mice, mirroring the design used in the JIMT-1 model as detailed above (Appendix A and Figure 5a–c).

Our experiments revealed that both second and third generation HER2-specific CAR NK cells efficiently recognized and killed the CD44^−^ N87 xenografts (Figure 5b, NT vs. all other treatments: *** *p* < 0.001), and there was no significant difference in tumor suppression between the CAR NK cells and the trastuzumab-redirected NK cells (Figure 5b, NT + TRAST vs. all HER2-CAR NK: n.s.). Although both treatments significantly prolonged survival (Figure 5c, NT vs. all treatments: *** *p* < 0.001), neither could completely eradicate the tumors, underlining the potential of CAR NK cells in targeted cancer therapy but also the need for further improvements to achieve better therapeutic outcomes.

## 4. Discussion

The development and clinical implementation of chimeric antigen receptor (CAR)-modified T cell therapies represent a significant advancement in oncology, particularly in the context of tumors resistant to targeted antibody-based biological therapy [8]. However, despite their success in treating certain hematological malignancies, CAR T cell therapies encounter several significant limitations. They are often associated with life-threatening side effects, including cytokine release syndrome (CRS) and neurotoxicity, which can limit their usability and require intensive management [30]. The production process for CAR T cells is complex and costly because it involves genetically modifying a patient’s own cells. This personalized approach requires significant time and resources [31]. Finally, the use of allogeneic CAR T cells carries the risk of inducing GvHD, a serious complication that can be fatal in some cases. This limits the potential for “off-the-shelf” solutions using these cells [32]. Given these significant challenges with CAR T cell therapies, CAR NK cells emerge as a promising alternative, potentially addressing each of these issues with innovative solutions that enhance safety, reduce production costs, and expand the scope of treatable cancers. CAR NK cells are associated with a reduced probability of causing severe side effects like CRS and neurotoxicity, presenting a safer alternative for cancer therapy [33]. Moreover, unlike CAR T cells, CAR NK cells can be derived from various sources, including cord blood and induced pluripotent stem cells. This allows for manufacturing “off-the-shelf” products that can be used in multiple patients, reducing production costs and improving accessibility [34]. Finally, CAR NK cells do not cause Graft versus Host Disease, making them suitable for allogeneic use. This attribute significantly broadens their therapeutic potential and safety [35].

Recent summaries of clinical trials using CAR NK cells for solid tumors highlight their promising potential but also underline significant challenges [15,16]. There is no definitive proof yet of their effectiveness against tumors that are resistant to monoclonal antibodies. The complex, CD44-rich tumor microenvironment of these tumors, adapted to evade standard antibody treatments, poses significant obstacles [3]. It remains uncertain whether CAR NK cells possess a mechanism similar to CAR T cells for identifying and eliminating these resistant tumors [8,9] or if they lack effectiveness, akin to antibody-dependent cellular cytotoxicity (ADCC), due to their inability to access antigens. Moreover, developing the optimal CAR design for NK cells—considering the selection of antigen targets, signaling domains, and costimulatory elements—remains a challenging task. The effectiveness of CAR NK cells depends heavily on these components, which are crucial for overcoming the adaptive resistance mechanisms in solid tumors.

Our study explores the application of HER2-specific CAR NK cells against both trastuzumab-sensitive (CD44^−^) and trastuzumab-resistant (CD44^+^) HER2-positive tumor models, providing insight into the potential and limitations of these innovative therapies. Our findings highlight a critical challenge in applying CAR NK cell therapies: the variability in treatment efficacy depending on the tumor microenvironment (TME). Specifically, the presence of CD44 in the ECM appears to be a pivotal factor in determining the responsiveness of tumor cells to CAR NK cells. In CD44-negative N87 models, both in vitro and in vivo experiments demonstrated significant tumor suppression and cytotoxicity by CAR NK cells. Conversely, in CD44-positive JIMT-1 models, CAR NK cells exhibited reduced efficacy, struggling to penetrate the ECM and reach tumor antigens effectively.

Beyond physical barriers, the TME is composed of various immune-suppressive cells, including regulatory T cells (Tregs), myeloid-derived suppressor cells (MDSCs), and tumor-associated macrophages (TAMs), all of which can inhibit T cell and NK-cell activity. While targeting a single immunosuppressive component, such as Tregs or MDSCs, may provide some therapeutic benefit, it is unlikely to be sufficient due to the redundancy and adaptability of immunosuppressive networks within the TME [36]. A more effective approach may involve combination strategies, such as engineering CAR NK or CAR T cells to secrete pro-inflammatory cytokines [37], incorporating checkpoint inhibitors [38], or co-administering agents that reprogram the TME to support antitumor immunity [39]. Additionally, designing CAR constructs that target multiple immune-suppressive pathways simultaneously could enhance treatment efficacy and prevent immune escape [40]. Our results suggest that understanding and manipulating the TME could significantly enhance the efficacy of CAR NK therapies. Therefore, future therapeutic developments must take a multi-targeted approach to overcoming TME-driven resistance, ensuring that immune effector cells maintain their cytotoxic function in hostile tumor settings.

The molecular backbone of CAR constructs, particularly the selection and combination of costimulatory endodomains, plays a crucial role in modulating the activity and persistence of CAR NK cells. In our study, constructs incorporating the 41BB costimulatory domain that have an immunomodulatory role in the NK cell activation demonstrated enhanced cytotoxic activity. This is consistent with the preclinical findings suggesting that 41BB promotes the survival and long-term persistence of CAR T cells [23,29]. These findings suggest that optimizing CAR design, primarily through the strategic use of costimulatory molecules, could significantly impact the outcome of CAR NK cell therapies.

The mixed efficacy observed in our study presents both a challenge and an opportunity for future research. For CAR NK cells to become a mainstay in cancer therapy, especially for ECM-rich and antibody-resistant tumors, developing strategies that enhance their penetration and functionality within hostile TMEs will be essential. Potential strategies could include the co-administration of ECM-degrading enzymes [41], the use of checkpoint inhibitors to reduce immunosuppression within the TME [42], or the engineering of CAR NK cells to secrete bispecific T cell engager molecules that could involve bystander tumor-infiltrating T cells into the antitumor response [25].

Our research contributes to the growing field of knowledge regarding CAR NK cell therapies and their application in complex oncological settings [43]. While the results are promising, they also highlight the need for a deeper understanding of TME dynamics and CAR construct optimization. By continuing to explore these areas, we can improve the design and efficacy of CAR NK cell therapies, ultimately enhancing their clinical utility in treating resistant forms of cancer.

## 5. Conclusions

Our study evaluates the efficacy of HER2-specific CAR NK cells against HER2-positive tumors with varying CD44 expression, providing insight into the potential and limitations of these therapies. While CAR NK cells showed considerable promise in the N87 CD44-negative, trastuzumab-sensitive tumor model, demonstrating effective tumor suppression both in vitro and in vivo, their performance was markedly less effective in trastuzumab-resistant, CD44-positive JIMT-1 tumors. The presence of CD44 in the extracellular matrix of these tumors creates a substantial barrier, impairing the CAR NK cells’ ability to reach and effectively target the tumor antigens. Our study emphasizes the need for strategies to overcome these barriers.

One strategy could be the optimization of the CAR constructs, possibly by incorporating specific costimulatory domains like 41BB that show potential for enhancing CAR NK cell activity and overcoming resistance mechanisms within challenging tumor microenvironments. This research contributes to the evolving understanding of CAR NK cell applications in oncology, suggesting that manipulating both the cellular components and the microenvironment could significantly improve therapeutic outcomes.

## Figures and Tables

**Figure 1 cancers-17-00731-f001:**
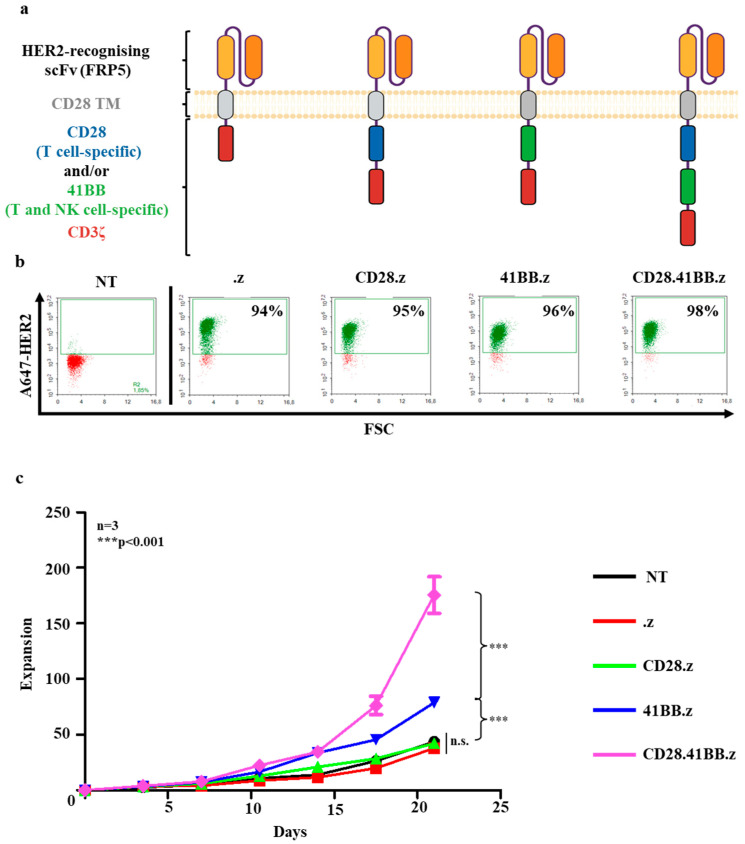
Generation of HER2-specific human CAR NK cells and their proliferation in vitro: (**a**) Schematic structure of HER2-targeting chimeric antigen receptor molecules (created with BioRender). (**b**) Representative flow cytometry dot-plots of non-transduced (NT) and HER2.z, HER2.CD28.z, HER2.41BB.z, and HER2.CD28.41BB.z CAR NK cells after FACS sorting. (**c**) Quantitative cell proliferation data. A total of 1 × 10^5^ HER2-CAR NK or NT NK cells were plated in duplicates in the presence of 400 IU/mL interleukin-2. Every 3.5 days, the effector cell number was determined by flow cytometry, and then the initial effector cell quantity was placed onto new plates in conditions identical to the beginning of the experiment. The expansion rate was the ratio of the cell number measured at the end and at the beginning of the last 3.5-day period. Histograms represent the mean ± SD (n = 3; assay in duplicates); *** *p* < 0.001.

**Figure 2 cancers-17-00731-f002:**
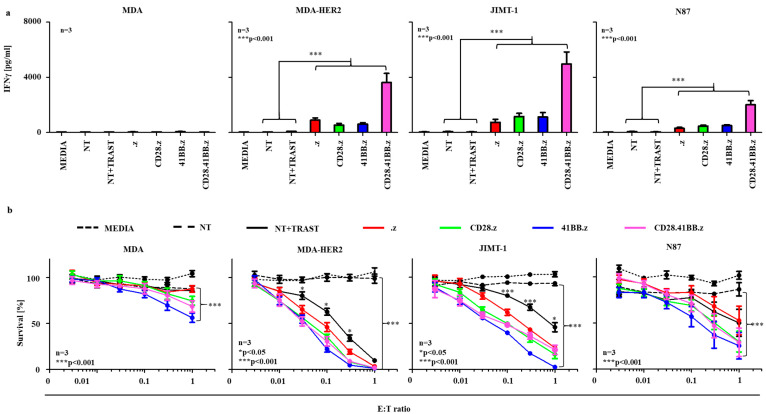
HER2-specific CAR NK cells successfully recognized and killed HER2-positive target cells: (**a**) 1 × 10^5^ HER2-CAR NK or NT NK cells ± 10 μg/mL trastuzumab were cocultured with MDA-HER2, JIMT-1, or N87 (HER2^+^) or MDA (HER2^−^) target cells in 1:1 NK cell-to-tumor cell ratio. After 24 h, IFNγ was determined in the culture supernatant by ELISA (n = 3, assay performed in duplicates). (**b**) Firefly-luciferase-based cytotoxicity assay using HER2-CAR NK or NT NK cells ± 10 μg/mL trastuzumab against 3 × 10^4^ MDA-HER2, JIMT-1, or N87 (HER2^+^) or MDA (HER2^−^) target cells at 1–0.3–0.1–0.03–0.01–0.003:1 NK cell-to-tumor cell ratio (marked as E:T ratio in the figure). Cell culture medium was not supplemented with IL-2 (n = 3; assay was performed in duplicates). Histograms show mean ± SEM; * *p* < 0.05; *** *p* < 0.001. All variants of HER2-CAR NK-92 cells were compared to the non-treated (NT control) at the effector-to-target ratio of 1:1. Additionally, the NT + TRAST group was compared to the .z group across the effector-to-target ratio range of 0.003 to 1.

**Figure 3 cancers-17-00731-f003:**
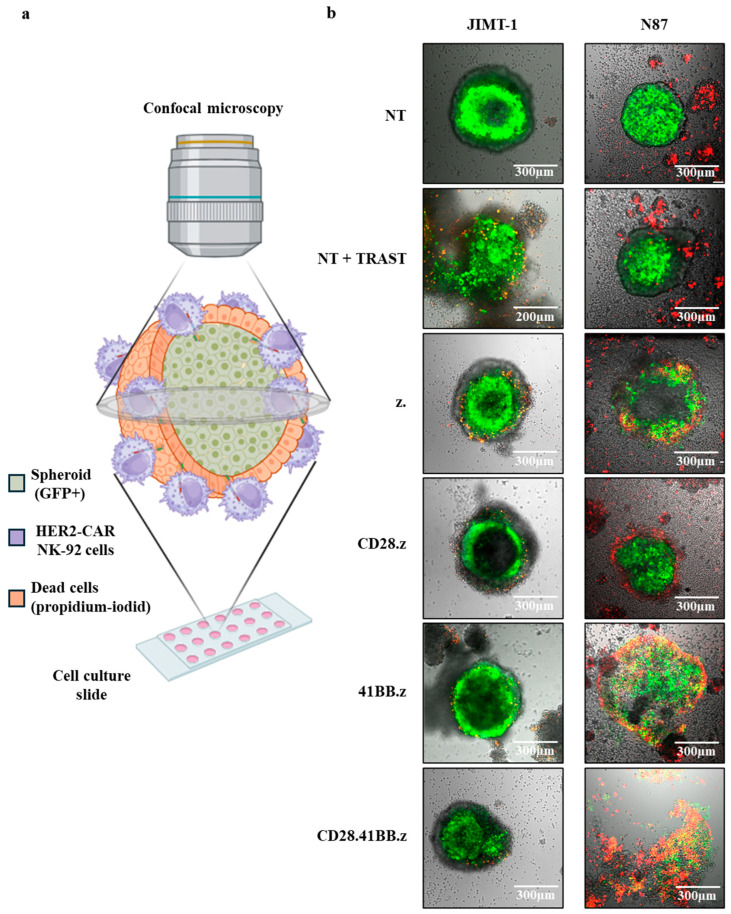
41BB-containing HER2-specific CAR NK cells selectively destroy three-dimensional CD44^−^ N87 tumor spheroids and spare CD44^+^ JIMT-1 spheroids: (**a**) Schematic representation of the optical sections through the spheroid. (**b**) Representative images (at 24 h) for detection of cytolytic activity of HER2-CAR NK cells and trastuzumab-targeted NT NK cells against HER2^+^/CD44^+^, trastuzumab-resistant JIMT-1 spheroids (left column), and HER2^+^/CD44^−^, trastuzumab-sensitive N87 spheroids (right column). Target cells are green (GFP expressing); dead cells are visualized by PI uptake (red). The actual scalebar is visible on the image and its dimensions are either 200 µm or 300 µm, depending on the specific image shown.

**Figure 4 cancers-17-00731-f004:**
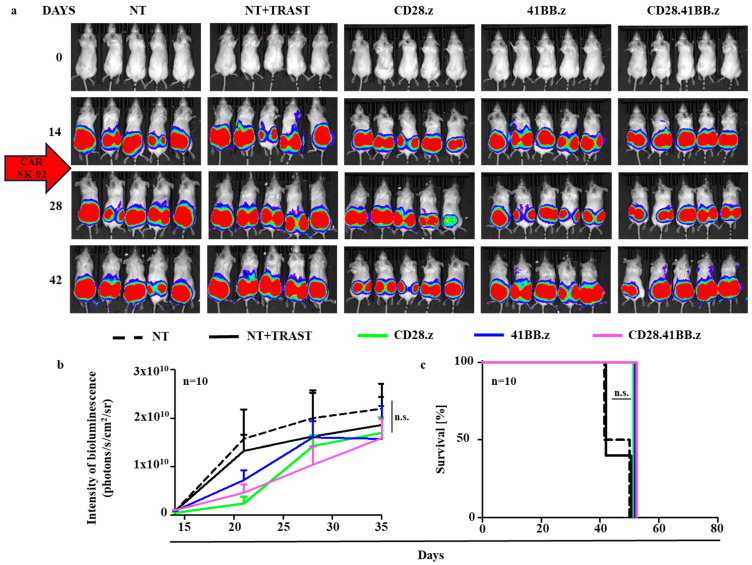
HER2-CAR NK cells are unable to eliminate CD44^+^, trastuzumab-resistant JIMT-1 xenografts in vivo. Mice were injected s.c. with 3 × 10^6^ JIMT-1.ffLuc cells. Mice on day 14 (arrow) received a single i.v. dose of 5 × 10^6^ NT NK cells (NT + TRAST; n = 10, solid black line and NT, n = 10, dashed black line) or HER2-CAR T cells (CD28.z, n = 10, green; 41BB.z, n = 10; blue and CD28.41BB.z, n = 10, pink). Tumor growth was followed by bioluminescence imaging. (**a**) Representative images of JIMT-1.ffLuc-injected animals. (**b**) Quantitative bioluminescence imaging data of JIMT-1.ffLuc xenografts (average total radiance = photons/s/cm^2^/sr). (**c**) Kaplan–Meier survival curve. Histograms represent mean ± SD.

**Figure 5 cancers-17-00731-f005:**
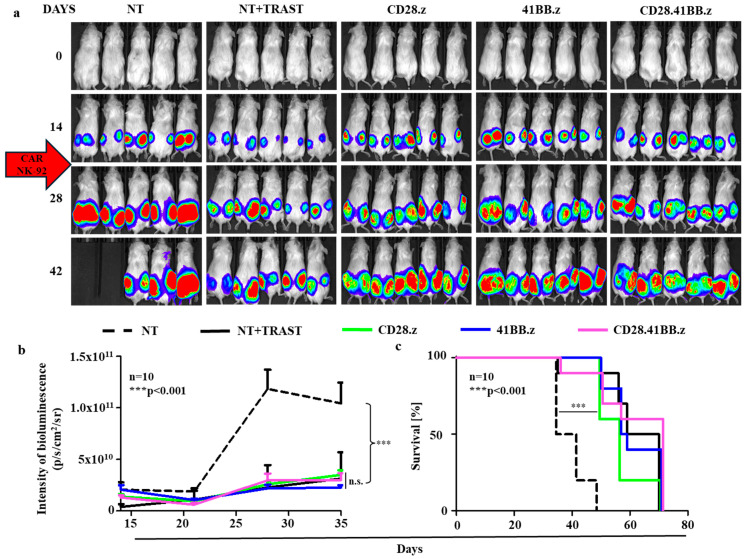
HER2-CAR NK cells exerted significant in vivo antitumor effect against CD44^−^. trastuzumab-sensitive N87 xenografts. Mice were injected s.c. with 3 × 10^6^ N87-1.ffLuc cells. Mice on day 14 (arrow) received a single i.v. dose of 5 × 10^6^ NT NK cells (NT + TRAST; n = 10, solid black line and NT, n = 10, dashed black line) or HER2-CAR T cells (CD28.z, n = 10, green; 41BB.z, n = 10, blue; and CD28.41BB.z, n = 10, pink). Tumor growth was followed by bioluminescence imaging. (**a**) Representative images of N87.ffLuc-injected animals. (**b**) Quantitative bioluminescence imaging data of JIMT-1.ffLuc xenografts (average total radiance = photons/s/cm^2^/sr) (NT vs. all other treatments: *** *p* < 0.001; NT + TRAST vs. HER2-CAR treatments: n.s.). (**c**) Kaplan–Meier survival curve (NT vs. all other treatments: *** *p* < 0.001; NT + TRAST vs. HER2-CAR treatments: n.s.). Histograms represent mean ± SD.

## Data Availability

The data presented in this study are available in this article and its Appendix A.

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
