# Peer review of "Targeting HER2-Positive Solid Tumors with CAR NK Cells: CD44 Expression Is a Critical Modulator of HER2-Specific CAR NK Cell Efficacy"

_cancers, 2025, doi:10.3390/cancers17050731_

Round 1

Reviewer 1 Report

Comments and Suggestions for Authors

Manuscript Revision ID: cancers-3478210

In the manuscript entitled “Targeting HER2-Positive Solid Tumors with CAR NK Cells: CD44 Expression Is a Critical Modulator of HER2-Specific CAR NK Cell Efficacy,” Gergely B. et al. examined the anti-tumor efficacy of the NK92 cell line genetically modified to express CARs targeting the HER2 antigen. They developed four HER2-CARs differing in costimulatory domain compositions. Interestingly, the authors demonstrate that the expression of CD44 on tumor target cells affects the efficacy of HER2-CAR NK cells in HER2+ cancer cell killing both in vitro and in vivo. HER2-CAR NK cells successfully eliminate both CD44+ and CD44- HER2+ target cells in monolayer cultures. Conversely, in a three-dimensional setting such as tumor spheroids and xenograft mouse models, the presence of the CD44 molecule on target cells inhibits the HER2-CAR NK cell anti-tumor effectiveness.

This study aims to point out advantages and, mostly, limitations of NK cell-based therapy for treating HER2+ solid tumors.

The manuscript is well written and scientifically pertinent. The rationale is sound, and the original hypothesis is tested using a properly designed experimental plan. The conclusions are justified and appropriately supported by the results.

I only have a few minor concerns:

  • Although the title reflects the content of the text and the main finding, it should be clearly stated that the study focuses on the NK92 cell line rather than primary NK cells, as one might understand.
  • 3. What effector-target ratio is utilized in the co-culture? Please indicate E:T in the figure legend or the corresponding result section.
  • 4, Fig. 5. These figures show that 25 mice in total received tumor cells subcutaneously. Then, as shown in panel A of both figures, mice were divided into 5 groups, with 5 animals per group. Therefore, why do the graphs in panels B and C of figures 4 and 5 indicate n=10? Please, clarify.
  • 4S. The authors indicated three replicates (n=3) in the figure legend, but the figure shows n=4. Furthermore, the figure legend indicates a *p<0.05, but this p-value is not represented in the actual figure. Please correct these inconsistencies.

Author Response

We would like to sincerely thank you for your valuable and insightful comments on our manuscript. Your constructive feedback has significantly contributed to improving the quality of our work. We have carefully addressed each of your comments and made the necessary modifications to the manuscript accordingly. Below is a point-by-point response detailing the changes made:

Reviewer's comment 1:
Although the title reflects the content of the text and the main finding, it should be clearly stated that the study focuses on the NK92 cell line rather than primary NK cells, as one might understand.

Author's reply:
We sincerely appreciate your thoughtful suggestion regarding the title. We understand the importance of ensuring clarity in conveying the focus of our study. However, after careful consideration, we believe that the current title effectively represents the scope and key findings of our research. In scientific literature, the NK92 cell line is widely utilized as a model for NK cell function, and many studies follow a similar approach in their titling without explicitly differentiating between cell lines and primary NK cells. To prevent any potential misunderstanding, we have made sure that the abstract, methods, and results sections explicitly state our use of NK92 cells. This ensures that readers can accurately interpret the scope of the study while maintaining consistency with standard conventions in the field. We hope this explanation addresses your concern and appreciate your understanding on this matter.

Reviewer's comment 2: 
What effector-target ratio is utilized in the co-culture? Please indicate E:T in the figure legend or the corresponding result section.

Author's reply: 
We appreciate your observation regarding the effector-target (E:T) ratio used in the co-culture. To clarify, we have now explicitly indicated the E:T ratio in the figure legends to ensure clarity for the readers.

Reviewer comment 3:
These figures show that 25 mice in total received tumor cells subcutaneously. Then, as shown in panel A of both figures, mice were divided into 5 groups, with 5 animals per group. Therefore, why do the graphs in panels B and C of figures 4 and 5 indicate n=10? Please, clarify.

Author's reply:
The total number of mice that received tumor cells subcutaneously was indeed 25, and they were divided into 5 groups, with 5 animals per group. However, each animal received 2 tumors, meaning that the total number of tumors per group was 10. In panels B and C of figures 4 and 5, n=10 refers to the number of tumors, not the number of animals. Since each mouse carried two tumors, statistical analyses were performed based on the total number of tumors rather than the number of individual mice. This approach enhances statistical power while reducing the total number of animals used, aligning with ethical considerations in preclinical research.

Reviewer comment 4:
The authors indicated three replicates (n=3) in the figure legend, but the figure shows n=4. Furthermore, the figure legend indicates a *p<0.05, but this p-value is not represented in the actual figure. Please correct these inconsistencies.

Author's reply:
Thank you for pointing this out! The n=4 in the figure represents the number of independent experiments, while in each experiment, we used 3 technical replicates. The p<0.05 indication was mistakenly delayed and is not reflected in the actual figure. We appreciate you catching these inconsistencies, and we corrected it accordingly.

We truly appreciate your time and effort to review our manuscript. Your constructive feedback has helped us to enhance the clarity and quality of our work significantly. We hope that the revised version meets your expectations.

Thank you once again for your insightful and valuable comments.

Reviewer 2 Report

Comments and Suggestions for Authors

Dear Authors,

This is a very good manuscript and should be published after some minor revisions. I purpose to write something about the possibility of targeting the tumorenvironment in the future. The main problem is that many different immune cells are able to inhibit the function of the T-cells and, is it enough to target only one specific cell ? maybe you can citate: Hanahan D, Michielin O, Pittet MJ. Convergent inducers and effectors of T cell paralysis in the tumour microenvironment. Nat Rev Cancer. 2025 Jan;25(1):41-58. doi: 10.1038/s41568-024-00761-z. Epub 2024 Oct 24. PMID: 39448877. 

The excellent figure on page 2 should be numbered Fig. 6 and inserted before the discussion with a title and legend. This summary would be very helpful for the readers. 

Author Response

We sincerely appreciate your positive feedback and constructive suggestions for improving our manuscript. Below, we provide our responses to your comments and outline the revisions accordingly. Our point-by-point replies:

Reviewer's question 1: I propose to write something about the possibility of targeting the tumor environment in the future. The main problem is that many different immune cells are able to inhibit the function of the T-cells and, is it enough to target only one specific cell ? maybe you can citate: Hanahan D, Michielin O, Pittet MJ. Convergent inducers and effectors of T cell paralysis in the tumour microenvironment. Nat Rev Cancer. 2025 Jan;25(1):41-58. doi: 10.1038/s41568-024-00761-z. Epub 2024 Oct 24. PMID: 39448877. 

Author's reply: 
Thank you for your recommendation to address the potential of targeting the TME in future therapies. We have incorporated a new paragraph in the discussion section elaborating on the complexity of the TME, particularly the presence of multiple immune-suppressive cells such as regulatory T cells (Tregs), myeloid-derived suppressor cells (MDSCs), and tumor-associated macrophages (TAMs). In this section, we discuss the challenges of targeting a single immunosuppressive component and propose alternative multi-targeted approaches, such as engineering CAR NK or CAR T cells to secrete pro-inflammatory cytokines, incorporating checkpoint inhibitors, or co-administering agents that reprogram the TME to enhance antitumor immunity. As per your suggestion, we have cited the relevant reference.

Added new paragraph in the Discussion: 
"Beyond physical barriers, the TME is composed of various immune-suppressive cells, including regulatory T cells (Tregs), myeloid-derived suppressor cells (MDSCs), and tumor-associated macrophages (TAMs), all of which can inhibit T-cell and NK-cell activity. While targeting a single immunosuppressive component, such as Tregs or MDSCs, may provide some therapeutic benefit, it is unlikely to be sufficient due to the redundancy and adaptability of immunosuppressive networks within the TME [34]. A more effective approach may involve combination strategies, such as engineering CAR NK or CAR T cells to secrete pro-inflammatory cytokines [35], incorporating checkpoint inhibitors [36], or co-administering agents that reprogram the TME to support antitumor immunity [37]. Additionally, designing CAR constructs that target multiple immune-suppressive pathways simultaneously could enhance treatment efficacy and prevent immune escape [38]. Our results suggest that understanding and manipulating the TME could significantly enhance the efficacy of CAR NK therapies. Therefore, future therapeutic developments must take a multi-targeted approach to overcoming TME-driven resistance, ensuring that immune effector cells maintain their cytotoxic function in hostile tumor settings."

Reviewer's question 2: 
The excellent figure on page 2 should be numbered Fig. 6 and inserted before the discussion with a title and legend. This summary would be very helpful for the readers. 

Author's reply:
We acknowledge your suggestion regarding the figure numbering and placement. However, based on the specific guidelines of the journal, we are unable to modify the numbering or relocate the figure within the manuscript. We appreciate your feedback and understand the value of improving readability, but we must adhere to the prescribed formatting requirements.

We truly appreciate your insightful comments, which have helped us refine our manuscript. Thank you for your time and consideration.

Reviewer 3 Report

Comments and Suggestions for Authors

The manuscript of Bence Gergely, et al. is devoted to the topic of antitumor application of natural killer cells expressing chimeric antigen receptor (CAR NK cells) for treatment solid tumors resistant to monoclonal antibody therapies. Based on their previous work authors used for this investigation in vitro and in vivo model of HER2-positive tumors and “generated NK-92 cell lines expressing 1st, 2nd and 3rd generation HER2-specific CARs with CD28 and/or 41BB costimulatory domains” – HER2-CAR NK-92 cells. It is important that in the work authors used two types of target tumor cells – resistant and sensitive to conventional monoclonal antibody therapies. This field of investigation is important for biomedicine, because the use of NK cells and development of CAR NK cells for antitumor therapy and the study of the mechanisms of their action is undoubtedly an urgent task.

In this work the authors demonstrated that CAR NK cells were effective against both monoclonal antibody-sensitive and -resistant tumors in monolayer cultures. “However, in three-dimensional spheroid models and in vivo xenografts they were less effective against CD44+ trastuzumab-resistant tumors. This reduced efficacy highlights the significant role of the tumor microenvironment, particularly the extracellular matrix, in hindering the therapeutic potential of CAR NK cells. Despite the promising in vitro performance of CAR NK cells, this study emphasizes the need for improved strategies to enhance their penetration and effectiveness in resistant tumors: optimizing CAR constructs and devising methods to overcome extracellular matrix barriers are crucial for advancing CAR NK cell therapies in oncology”. I have nothing to add to this conclusion made by the authors of the manuscript about their work. It testifies to the productivity of the research conducted, despite the absence of the expected result. There is no doubt that the data presented in the manuscript will be useful for developing methods for the practical use of CAR NK cells in oncotherapy.

In general the study is well presented, proper controls are used and the conclusions are convincingly supported by experimental results, the data are of considerable novelty and interest. Manuscript is well written. In fact, the manuscript is ready for publication, but several minor suggestions might improve the overall quality of the manuscript:

  1. (line 75) The abbreviations GvHD should be expanded.
  2. (line 109) The abbreviation ECM should be expanded.
  3. (line 220) The abbreviation i.p. should be expanded.
  4. (line 224) The abbreviations i.v. should be expanded.

Author Response

We sincerely appreciate your thorough and positive review of our manuscript. Your kind words and thoughtful evaluation of our work mean a lot to us, and we are grateful for your support and constructive suggestions. It is truly encouraging to hear that you find our study relevant and important for advancing CAR NK cell therapies in oncology.

We are delighted that you acknowledge the significance of our findings and the challenges presented by the tumor microenvironment. As you pointed out, the reduced efficacy of CAR NK cells in 3D models and in vivo xenografts underscores the critical need for improved strategies to enhance their penetration and therapeutic potential. We hope that our research contributes to the development of more effective CAR NK cell-based treatments for resistant tumors.

Revisions made by your suggestions: 
We have now ensured that all abbreviations, including GvHD (graft-versus-host disease), ECM (extracellular matrix), i.p. (intraperitoneal), and i.v. (intravenous), are properly expanded at their first appearance in the main text. Additionally, we have included them in the Abbreviation section for further clarity.

Once again, we sincerely appreciate your invaluable feedback, and we are grateful for your support in refining our manuscript.